# Frailty predicts surgical complications after kidney transplantation. A propensity score matched study

**Milena dos Santos Mantovani[1], Nyara Coelho de Carvalho[1], Thomáz Eduardo Archangelo[1], Luis Gustavo Modelli de Andrade[2], Sebastião Pires Ferreira Filho[3], Ricardo de Souza Cavalcante[3], Paulo Roberto Kawano[4], Silvia Justina Papini[5], Nara Aline Costa[1], Ricardo Augusto Monteiro de Barros Almeida[3]***

**1** Faculdade de Medicina de Botucatu, Universidade Estadual Paulista (Unesp), Botucatu, São Paulo, Brasil, **2** Departamento de Clínica Médica, Faculdade de Medicina de Botucatu, Universidade Estadual Paulista (Unesp), Botucatu, São Paulo, Brasil, **3** Departamento de Doenças Tropicais e Diagnóstico por Imagem, Faculdade de Medicina de Botucatu, Universidade Estadual Paulista (Unesp), Botucatu, São Paulo, Brasil, **4** Departamento de Urologia, Faculdade de Medicina de Botucatu, Universidade Estadual Paulista (Unesp), Botucatu, São Paulo, Brasil, **5** Departamento de Enfermagem, Faculdade de Medicina de Botucatu, Universidade Estadual Paulista (Unesp), Botucatu, São Paulo, Brasil

* almeidaramb@yahoo.com.br

## Abstract

### Background and objective

Surgical complications after kidney transplantation can lead to catastrophic outcomes. Frailty has been associated with important kidney transplantation outcomes; however, there are no studies assessing this measure of physiological reserve as a specific predictor of surgical complications in this population. Such an assessment was, therefore, the objective of the present study.

### Methods

A total of 87 individuals aged ≥ 18 years who underwent kidney transplantation between March 2017 and March 2018 were included. At the time of admission for kidney transplantation, demographic, clinical, and kidney transplantation data were collected, and the frailty score was calculated according to Fried et al., which comprises five components: shrinking, weakness, exhaustion, low activity, and slowed walking speed. Urological, vascular, and general surgical complications were assessed three months later, or until graft loss or death. The propensity score was used to achieve a better homogeneity of the sample, and new analyses were performed in this new, balanced sample.

### Results

Of the 87 individuals included, 30 (34.5%) had surgical complications. After propensity score matching, the risk of surgical complications was significantly higher among the frail individuals (RR 2.14; 95% CI 1.01–4.54; p = 0.035); specifically, the risk of noninfectious

**Data Availability Statement:** All relevant data are within the manuscript and its Supporting Information files.

**Funding:** This study was supported by grant from São Paulo Research Foundation (FAPESP) # 2016/24745-3 (http://www.fapesp.br/en/) and was also financed in part by the Coordenação de Aperfeiçoamento de Pessoal de Nível Superior - Brasil (CAPES) - Finance Code 001 (https://www.capes.gov.br/). TEA received a scientific initiation scholarship from the National Council for Scientific and Technological Development (CNPq), RT - Bolsa Reitoria, project 42014 (http://www.cnpq.br/).

**Competing interests:** The authors have declared that no competing interests exist.

surgical complications was significantly higher among these individuals (RR 2.50; 95% CI 1.11–5.62; p = 0.017).

## Conclusion

The results showed that individuals with some degree of frailty before kidney transplantation were more subject to surgical complications. The calculation of the frailty score for transplant candidates and the implementations of measures to increase the physiological reserve of these patients at the time of kidney transplantation may possibly reduce the occurrence of surgical complications.

## Introduction

Despite advances in surgical techniques and the use of new technologies, kidney transplantation (KTx) is still associated with various clinical and surgical complications due to the high complexity of this procedure [1–3]. Although the overall incidence of surgical complications is relatively low in KTx, especially when compared to other organs such as the liver or pancreas, they are usually present in approximately 2.5–15% of cases and, if not diagnosed and treated properly, can lead to catastrophic outcomes [3–5].

Although several classifications have been proposed, surgical complications can typically be divided into urological and vascular complications. The most common urological complications, usually present in up to 15% of patients, are urinary leak, ureteral obstruction/stricture, lithiasis, and vesicoureteral reflux, whose treatments will depend on the time of onset and severity of the condition, among other variables [4,6,7]. In turn, vascular complications, observed in 3 to 15% of cases, tend to have less favorable outcomes. With the exception of lymphocele and renal artery stenosis, pseudoaneurysms and vascular thromboses (of either the renal artery or vein) typically progress to graft loss, regardless of the diagnosis and/or applied treatment [8–10]. Other complications of KTx can be classified as general complications, and these involve mainly surgical wound dehiscence/infection [3,4].

The identification of predictors of outcomes in the kidney-transplanted population is essential, aiming to more adequately guide the inclusion and maintenance of patients on the waiting list and to enable the most adequate control of these predictor factors before KTx. However, most models studied have little effectiveness in predicting the most relevant outcomes of KTx [11,12].

Frailty is a measure of physiological reserve, initially validated for the geriatric population [13]. Although the frailty score has not been formally validated for patients with end-stage renal disease (ESRD) and for kidney-transplanted patients, it has been shown to be applicable to these populations. These patients appear to share many pathogenic mechanisms of frailty, such as a pro-inflammatory state, with an exacerbated production of inflammatory cytokines, and dysregulation of the immune, neuroendocrine, and neuromuscular systems, resulting in accelerated ageing [12,14–16]. Frailty is considered highly prevalent in patients at any stage of chronic kidney disease (CKD) and may reach up to two-thirds in ESRD cases [17]. The use of the frailty score has progressively increased in the ESRD and transplant population. A recent systematic review [18] evaluating the use of the frailty score in the population of patients with CKD found that 72% of the studies used the Fried et al. score [13].

Frailty has been studied as a predictor of KTx outcomes. To date, frailty has been shown to be an independent risk factor for delayed graft function [12], post-transplant delirium [19],

longer initial length of hospital stay [20], early hospital readmission [21], adverse effects from immunosuppressive drugs [22], and death [23]. Importantly, the studies that have been conducted thus far are limited to only one transplant center in the United States.

Studies have shown an association between frailty and general postoperative complications related to KTx [24] and to other types of surgery [25–29], but to the best of our knowledge, no studies have evaluated frailty as a specific predictor of surgical complications after KTx.

The relevance of surgical complications, the need to identify their predictive factors, and the need for scientific evidence regarding the efficacy of frailty as a good predictor of KTx outcomes justified the present study.

## Materials and methods

### Study design

A prospective and longitudinal study was conducted that included 87 KTx recipients whose transplants were performed by the Kidney Transplant Program of Botucatu Medical School Clinics Hospital, located in the city of Botucatu, São Paulo, Brazil. Botucatu Medical School Clinics Hospital is a tertiary-level care, education and research center with 417 beds, covering approximately 75 municipalities and 2 million people. Annually, approximately 140 KTx procedures are performed in this center, with 80% of these using deceased donors.

The study included individuals aged ≥ 18 years who underwent KTx between March 2017 and March 2018. Individuals undergoing transplantation of other organs along with the kidney were excluded. Individuals with amputations or other physical conditions that prevented the walk test or the handgrip strength test and the patients with significant cognitive impairment who were unable to understand and respond to the frailty score questionnaires were also excluded from the study.

All participants were evaluated at admission (M0) and at three months after the KTx procedure, or until graft loss or death (M1). At time M0, demographic, clinical, and KTx data were collected, including age, sex, race/ethnicity, etiology of chronic kidney disease, comorbidities, body mass index (BMI), cardiovascular risk, type of dialysis and time on dialysis, number of previous kidney transplants, type of donor, donor data, compatibility and immunological risk, cold ischemia time, and immunosuppressive regimens, and the frailty score was calculated according to Fried et al. [13], as described below. Data related to surgical complication outcomes were collected at time M1.

The data above were collected from electronic medical records and from interviews with the patients.

### Surgical protocol, prophylaxis against surgical site infections, and prevention of vascular complications of KTx

The surgical protocol, prophylaxis against surgical site infections, and the vascular complication prevention regimen used by the KTx unit are described in detail in the (S1 Appendix).

### Frailty

The frailty score used was based on Fried et al. [13], and was validated for the Brazilian population [30,31]. The score is divided into five components: shrinking (self-report of unintentional weight loss ≥ 10 lb in the previous year and/or BMI < 18.5 kg/m$^2$); weakness (using a handheld dynamometer, considering sex and BMI); feeling of exhaustion (self-reported); slow walking speed (time taken to walk 15 ft, considering height and sex); and low weekly physical activity, considering the patient's sex. Fried et al. [13] score was calculated considering only the dry

weight loss. More detailed descriptions of the five components of the frailty score can be found in the (S2 Appendix).

The frailty score is calculated by summing the scores of the components, with each component corresponding to one point, where 1 corresponds to the presence of the component and 0 corresponds to its absence. A score of 0 to 1 indicates the absence of frailty, 2 indicates intermediate frailty, and 3 to 5 indicate frailty.

## Statistical analysis

Sample size was estimated based on the study of Araújo et al. [32]. We assumed a frequency of surgical complications of 45% in the frail group and 15% in the non-frail group. Considering a power of 0.80, and alpha of 0.05, we calculated a minimum sample size of 68 patients [33].

Univariate analysis was performed using the variables of the recipient and donor, grouped by the presence of frailty. Frailty was divided into two categories: non-frail and frail (the latter grouping intermediate frailty and frailty). The t-test was used for the parametric continuous variables, and the Mann-Whitney test was used for the nonparametric variables. For the categorical variables, the chi-squared test or Fisher's exact test were used, as appropriate.

The propensity score was used to achieve a better homogeneity of the sample. The variables related to the presence or absence of surgical complications were used to calculate the propensity score using the logistic regression method. The variables selected for this score were age, sex, presence of diabetes, time on dialysis before KTx, cardiovascular risk, BMI classification, panel reactive antibodies, type of donor, expanded criteria donor, and cold ischemia time. After calculating the propensity score, the group selection (matching) method used was the k-nearest neighbor (KNN) method, with a caliper width of 0.2. A visual analysis of the normal quantile-quantile (QQ) plots was performed to confirm that the groups were adequately balanced. The statistical analyses were performed on 'before matching' and 'after matching'. The analyses after matching were performed using the balanced sample.

All analyses were performed using R (R Foundation for Statistical Computing, Vienna, Austria) version 3.4.2 with the MatchIt package to match and package pwr to sample size (S3 and S4 Appendices).

We considered statistical significance p value < 0.05. All analyses considered two-tailed hypothesis test.

## Ethics in research

The present study was approved by the Research Ethics Committee of Botucatu Medical School (protocol number 1.782.708).

## Results

### Study population

A total of 131 KTx procedures were performed between March 2017 and March 2018 at the Botucatu Medical School Clinics Hospital. Of these, 87 (66.4%) patients were included in the study. The reasons for not including the other 44 patients are shown in Fig 1. The non-inclusion of patients due to logistical difficulties occurred mainly due to communication difficulties between transplant and research teams at the time of admission for kidney transplantation.

### Baseline characteristics

Table 1 shows the general characteristics of the 87 patients included in the study and of the deceased donors, with a group of 32 (36.8%) individuals considered frail [frail (n = 14; 16.1%)

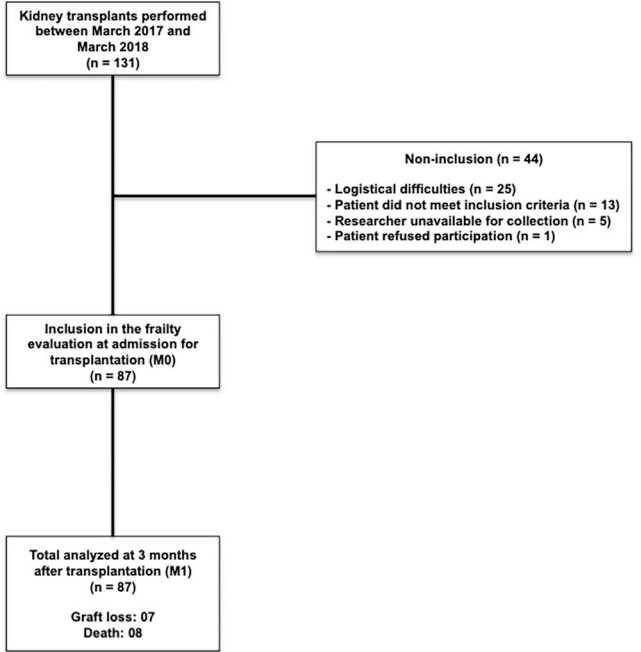

**Fig 1. Patient disposition.**

or with intermediate frailty (n = 18; 20.7%)] and a group of 55 (63.2%) non-frail individuals, before and after matching.

Before matching, there was a significant difference between the frail and non-frail groups only with respect to sex, with female patients being considered more frail, and regarding panel reactive antibodies, whose levels were higher in the frail group. After matching, there was no significant difference for the study variables between the two groups. The post-matching analysis showed a homogeneous distribution of the propensity scores between 0.2 and 0.8, with the outliers being removed (S5 Appendix).

No patient had a serological test positive for human immunodeficiency virus (HIV), a positive PCR result for hepatitis C virus (HCV), or the presence of hepatitis B surface antigen (HBsAg).

## Frailty and surgical complications

Of the 87 individuals included, 30 (34.5%) had surgical complications during the evaluated period—four (4.6%) patients developed infectious complications and 29 (33.3%) developed noninfectious complications. Table 2 lists the surgical complications identified in the study.

Table 3, Figs 2 and 3 show the occurrence of surgical complications after KTx, according to the frail and non-frail groups, before and after matching. Before matching, the risk of noninfectious surgical complications was significantly higher among the frail individuals (RR 1.84; 95% CI 1.03–3.30; p = 0.041). After matching, according to the propensity score, the risk of surgical complications was significantly higher among the frail individuals (RR 2.14; 95% CI 1.01–4.54; p = 0.035); specifically, the risk of noninfectious surgical complications was significantly higher among these individuals (RR 2.50; 95% CI 1.11–5.62; p = 0.017).

The incidence of graft loss (3.1% vs. 15.6%, p = 0.086) and death (9.4% vs. 12.5%, p = 0.689) were not statistically different after matching, respectively, between non-frail and frail groups.

**Table 1. Patient and donor characteristics.**

| Patient characteristics | Before matching | | | After matching | | |
|---|---|---|---|---|---|---|
| | Non-frail (n = 55) | Frail (n = 32) | P | Non-frail (n = 32) | Frail (n = 32) | P |
| Age, mean (SD), y | 44 (12) | 46 (13) | 0.519[1] | 48 (12) | 46 (13) | 0.535[1] |
| Male (%) | 67.3 | 43.8 | 0.032[2] | 56.2 | 43.8 | 0.317[2] |
| Caucasian (%) | 47.3 | 59.4 | 0.276[2] | 43.8 | 59.4 | 0.211[2] |
| Cause of ESRD (%) | | | | | | |
| Diabetes mellitus | 9.1 | 12.5 | 0.904[2] | 12.5 | 12.5 | 0.846[2] |
| Hypertension | 23.6 | 18.8 | | 25.0 | 18.8 | |
| Glomerulonephritis | 5.5 | 9.4 | | 3.1 | 9.4 | |
| Undefined | 32.7 | 34.4 | | 31.2 | 34.4 | |
| Other | 29.1 | 25.0 | | 28.1 | 25.0 | |
| Retransplantation (%) | 3.6 | 3.1 | 1.000[3] | 3.1 | 3.1 | 1.000[3] |
| BMI classification (%) | | | | | | |
| Underweight | 3.6 | 9.4 | 0.400[2] | 3.1 | 9.4 | 0.190[2] |
| Normal weight + overweight | 80.0 | 68.8 | | 87.5 | 68.8 | |
| Obesity | 16.4 | 21.9 | | 9.4 | 21.9 | |
| Cardiovascular risk (%) | | | | | | |
| Low | 74.5 | 62.5 | 0.237[2] | 62.5 | 62.5 | 1.000[2] |
| Moderate + High | 25.5 | 37.5 | | 37.5 | 37.5 | |
| Pre-KTx diabetes (%) | 12.7 | 15.6 | 0.705[2] | 15.6 | 15.6 | 1.000[2] |
| Time on dialysis, median [IQR], m | 26 [12–41] | 29 [21–57] | 0.265[4] | 31 [15–49] | 29 [21–57] | 0.742[4] |
| Dialysis modality (%) | | | | | | |
| None (preemptive) | 3.6 | 3.1 | 0.214[2] | 3.1 | 3.1 | 0.484[2] |
| Hemodialysis | 76.4 | 90.6 | | 81.2 | 90.6 | |
| Peritoneal | 20.0 | 6.2 | | 15.6 | 6.2 | |
| PRA, median [IQR], % | 0 [0–0] | 0 [0–67] | 0.030[4] | 0 [0–0] | 0 [0–67] | 0.089[4] |
| HLA mismatches, median [IQR], n | 3 [2–4] | 3 [3–3] | 0.497[4] | 3 [2–4] | 3 [3–3] | 0.916[4] |
| Deceased donor (%) | 74.5 | 84.4 | 0.285[2] | 71.9 | 84.4 | 0.226[2] |
| Induction therapy (%) | | | | | | |
| No induction | 3.6 | 3.1 | 0.900[2] | 6.2 | 3.1 | 0.554[2] |
| Anti-thymocyte globulin | 96.4 | 96.9 | | 93.8 | 96.9 | |
| Maintenance therapy (%) | | | | | | |
| FK+mTORi+PDN* | 65.5 | 59.4 | 0.670[2] | 62.5 | 59.4 | 0.763[2] |
| FK+MPS+PDN | 29.1 | 37.5 | | 31.2 | 37.5 | |
| Other | 5.5 | 3.1 | | 6.2 | 3.1 | |
| Cold ischemia time, median [IQR], h | 21 [8–24] | 23 [20–26] | 0.107[4] | 21 [2–24] | 23 [20–26] | 0.145[4] |
| **Deceased donor characteristics** | | | | | | |
| Age, mean (SD), y | 37 (14) | 43 (15) | 0.077[1] | 38 (15) | 43 (15) | 0.151[1] |
| Cause of death (%) | | | | | | |
| Stroke | 48.8 | 59.3 | 0.541[2] | 52.2 | 59.3 | 0.476[2] |
| Traumatic brain injury | 41.5 | 37.0 | | 34.8 | 37.0 | |
| Other | 9.8 | 3.7 | | 13.0 | 3.7 | |
| Serum creatinine median [IQR], mg/dL | 1.0 [0.8–1.2] | 1.0 [0.8–1.3] | 0.831[4] | 1.0 [0.8–1.0] | 1.0 [0.8–1.3] | 0.516[4] |
| Diabetes mellitus (%) | 3.6 | 9.4 | 0.267[3] | 0.0 | 9.4 | 0.076[3] |
| Hypertension (%) | 18.2 | 21.9 | 0.675[2] | 18.8 | 21.9 | 0.756[2] |

(*Continued*)

**Table 1.** (Continued)

| Patient characteristics | Before matching | | | After matching | | |
|---|---|---|---|---|---|---|
| | Non-frail (n = 55) | Frail (n = 32) | P | Non-frail (n = 32) | Frail (n = 32) | P |
| Expanded Criteria Donor (%) | 16.4 | 28.1 | 0.323[2] | 18.8 | 28.1 | 0.412[2] |

SD, standard deviation; ESRD, end-stage renal disease; BMI, body mass index; KTx, kidney transplantation; IQR, interquartile range; PRA, panel reactive antibody; HLA, Human leukocyte antigen; FK, tacrolimus; mTORi, mammalian target of rapamycin inhibitor; MPS, mycophenolate sodium; PDN, prednisone.

*Tacrolimus+everolimus+prednisone, 87.3%; tacrolimus+sirolimus+prednisone, 12.7%.

[1] t-test.

[2] Pearson´s Chi-squared test.

[3] Fisher's exact test.

[4] Mann-Whitney test.

**Table 2. Post-transplant surgical complications.**

| Surgical complications | n patients (%) |
|---|---|
| **Urological** | |
| Urinary leak | 7 (8.0) |
| Hydronephrosis | 2 (2.3) |
| Distal ureteral necrosis | 1 (1.1) |
| **Vascular** | |
| Renal artery stenosis | 1 (1.1) |
| Renal vein thrombosis | 4 (4.6) |
| Perigraft bleeding | 4 (4.6) |
| Perigraft hematoma | 5 (5.7) |
| Lymphocele | 5 (5.7) |
| Infected lymphocele | 1 (1.1) |
| **General** | |
| Abdominal wall seroma | 2 (2.3) |
| Abdominal wall hematoma | 2 (2.3) |
| Surgical wound dehiscence | 3 (3.4) |
| Superficial surgical site infection | 2 (2.3) |
| Abdominal wall infection | 1 (1.1) |
| Perigraft infection | 1 (1.1) |

**Table 3. Surgical complications after kidney transplantation, according to the frail and non-frail groups, before and after matching.**

| | Before matching | | | | After matching | | | |
|---|---|---|---|---|---|---|---|---|
| | Non-frail (n = 55) | Frail (n = 32) | P | RR (95% CI) | Non-frail (n = 32) | Frail (n = 32) | P | RR (95% CI) |
| Surgical complications n (%) | 15 (27.3) | 15 (46.9) | 0.064[1] | 1.72 (0.97–3.03) | 7 (21.9) | 15 (46.9) | 0.035[1] | 2.14 (1.01–4.54) |
| Infectious n (%) | 2 (3.6) | 2 (6.2) | 0.575[2] | 1.72 (0.25–11.62) | 2 (6.2) | 2 (6.2) | 1.000[2] | 1.00 (0.15–6.67) |
| Noninfectious n (%) | 14 (25.5) | 15 (46.9) | 0.041[1] | 1.84 (1.03–3.30) | 6 (18.8) | 15 (46.9) | 0.017[1] | 2.50 (1.11–5.62) |

RR: relative risk; CI: confidence interval.

[1] Pearson´s Chi-squared test.

[2] Fisher's exact test.

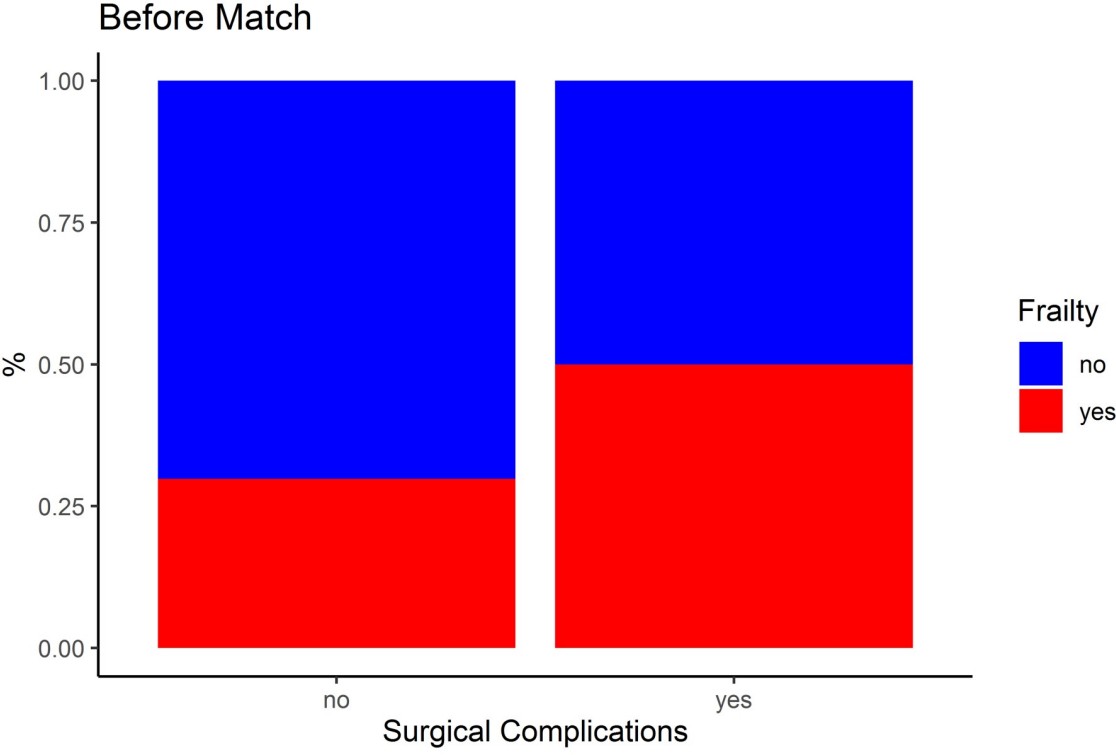

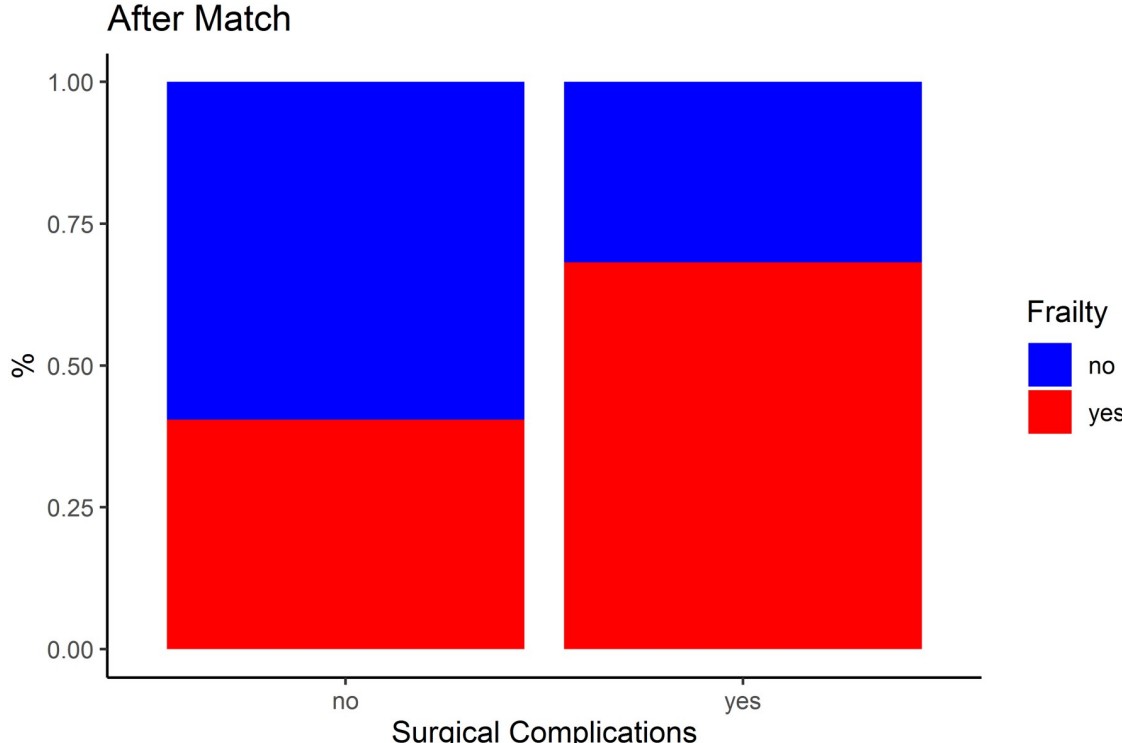

**Fig 2. Surgical complications after kidney transplantation, according to the frail and non-frail groups, before and after matching.**

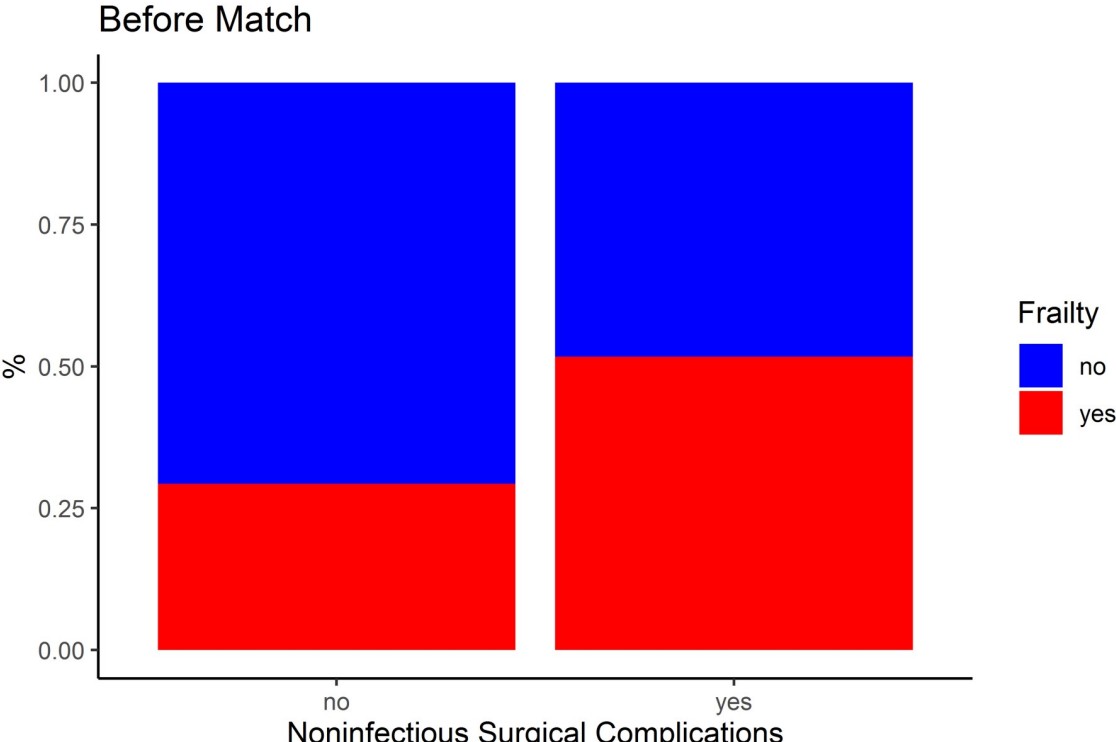

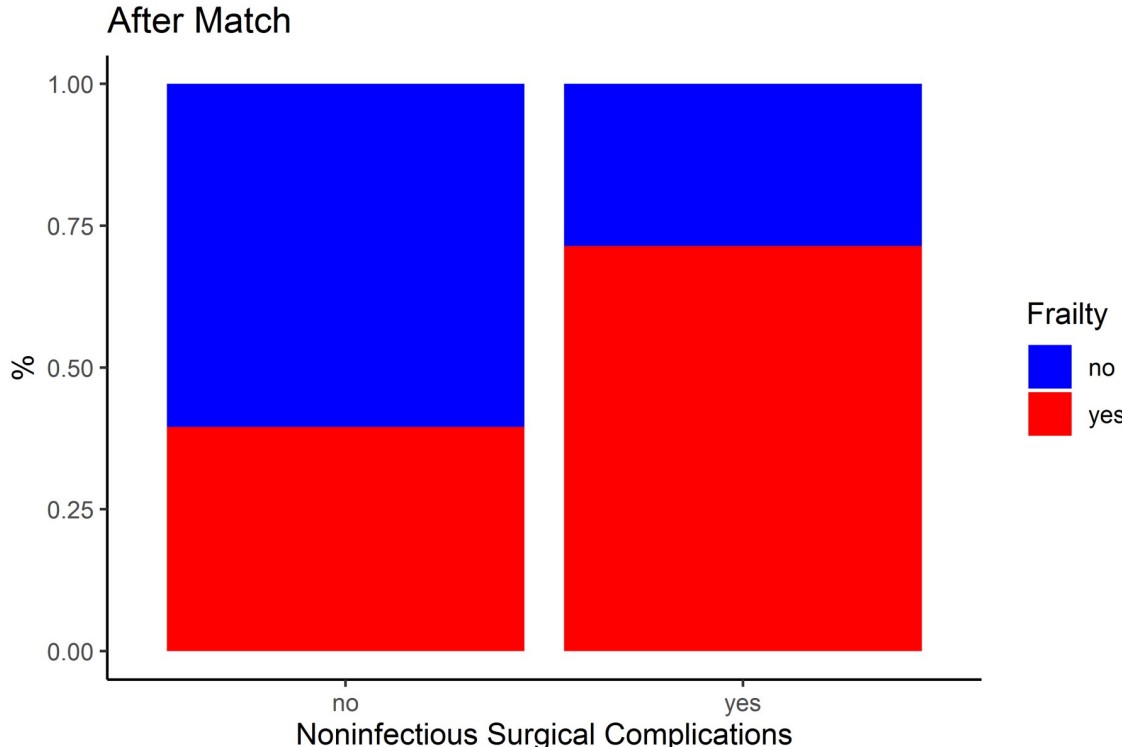

**Fig 3. Noninfectious surgical complications after kidney transplantation, according to the frail and non-frail groups, before and after matching.**

## Discussion

Until the present study, no studies evaluating the frailty score as a specific predictor of surgical complications after KTx have been identified.

In general, the frequency of urological, vascular, and general surgical complications in the studied sample was in agreement with the literature [3,4,6–10]. Urinary leak was the most frequent complication, accounting for 8.0% of the total. Also noteworthy is the low incidence of arterial stenosis observed in this study (1.1%), which is lower than that reported in the literature. Notably, mammalian target of rapamycin (mTOR) inhibitors with low tacrolimus levels was the most frequently used immunosuppressive maintenance regimen in the present study, and this combination has been associated with slightly higher risk of wound healing complications [34].

The percentage of frail individuals in the study sample (16.1%) was slightly lower compared to the literature. McAdams-DeMarco et al. [35] found 19.5% of frail individuals at admission for KTx using the Fried et al. score [13], whereas another multicenter study found 18.4% of frail individuals on the waiting list for KTx [36]. When considering the frequency of patients with intermediate frailty, the percentages found in these two studies increase to 31.7% [35] and 62.7% [36], respectively—much higher than that identified in the present study (20.7%). Differences among the studied populations, especially the higher age range of the KTx recipients included in the studies cited above, may explain such discrepant frequencies.

Fried et al. [13] found a higher prevalence of frailty among elderly women compared to elderly men in the same age group, corroborating the results of the present study. However, this difference was not detected in large-scale studies that evaluated populations that were more similar to the population studied here [35–37].

After matching, the risk of surgical complications, more specifically of noninfectious complications, was 2.5 times greater among individuals with some degree of frailty. Although this association may seem intuitive, this was the first time that this association was scientifically evidenced.

Studies have shown the association between frailty and general postoperative complications related to other types of surgery using specific population groups and various tools to measure frailty [25–29]. However, those studies selected a broader range of postoperative complications as outcomes, using classifications such as the one proposed by Clavien et al. or according to the criteria proposed by the American College of Surgeons National Surgical Quality Improvement Program (NSQIP) [29,38,39], and not specifically surgical complications, as evaluated for the first time in the present study.

Schopmeyer et al. [24] associated the occurrence of postoperative complications (30 days after KTx) with the frailty score calculated at admission for KTx using a different tool than that proposed by Fried et al. [13]. However, the study also evaluated general postoperative complications as the outcome using the Comprehensive Complication Index (CCI) [40], and not specifically surgical complications.

Because the present study evaluates an unprecedented outcome in the population of kidney-transplanted patients, it is not possible to compare its results with the literature.

The present study has relevant limitations. The collection of data from a single center makes it difficult to extrapolate the results to other populations with characteristics different from our center. Increasing the number of patients in the sample would be desired to better evaluate the association between frailty and the different categories of post-KTx surgical complications. The exclusion of patients due to logistical difficulties may have resulted in a systematic bias. However, we believe that this potential bias does not represent a relevant risk, as we cannot anticipate the frail condition without applying the Fried score [13].

However, the present study also has its strengths. This study is a prospective and unprecedented study. Although there is still no ideal method for calculating the frailty score in patients who undergo KTx [41], the tool of Fried et al. [13] is certainly the most used, and it was validated for the Brazilian population. Importantly, the frailty data was collected at patient admission, reflecting the patients' physiological reserve for KTx at the most appropriate time possible. Finally, it is also noteworthy that this is the first study evaluating the association between frailty and KTx in the Brazilian population.

The results of the present study showed that individuals with some degree of frailty before KTx were more subject to surgical complications, more specifically, to noninfectious complications. The calculation of the frailty score for transplant candidates and the implementations of measures to increase the physiological reserve of these patients at the time of KTx may possibly reduce the occurrence of surgical complications.

Studies evaluating the association between frailty and surgical complications should be performed in different populations of KTx recipients, using a larger sample.

## Supporting information

**S1 Appendix. Surgical protocol, prophylaxis against surgical site infections, and prevention of vascular complications in the kidney transplant unit.**
(PDF)

**S2 Appendix. Frailty score components.**
(PDF)

**S3 Appendix. Code R related to data and statistics.**
(R)

**S4 Appendix. Tabulated data.**
(XLSX)

**S5 Appendix. Distribution of propensity scores after matching.**
(PDF)

## Author Contributions

**Conceptualization:** Milena dos Santos Mantovani, Luis Gustavo Modelli de Andrade, Sebastião Pires Ferreira Filho, Ricardo de Souza Cavalcante, Silvia Justina Papini, Nara Aline Costa, Ricardo Augusto Monteiro de Barros Almeida.

**Data curation:** Milena dos Santos Mantovani, Luis Gustavo Modelli de Andrade, Ricardo Augusto Monteiro de Barros Almeida.

**Formal analysis:** Luis Gustavo Modelli de Andrade.

**Funding acquisition:** Ricardo Augusto Monteiro de Barros Almeida.

**Investigation:** Milena dos Santos Mantovani, Nyara Coelho de Carvalho, Thomáz Eduardo Archangelo, Sebastião Pires Ferreira Filho, Ricardo de Souza Cavalcante, Paulo Roberto Kawano, Ricardo Augusto Monteiro de Barros Almeida.

**Project administration:** Ricardo Augusto Monteiro de Barros Almeida.

**Writing – review & editing:** Milena dos Santos Mantovani, Nyara Coelho de Carvalho, Thomáz Eduardo Archangelo, Luis Gustavo Modelli de Andrade, Sebastião Pires Ferreira Filho,

Ricardo de Souza Cavalcante, Paulo Roberto Kawano, Silvia Justina Papini, Nara Aline Costa, Ricardo Augusto Monteiro de Barros Almeida.

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
