## [Decision Letter · Decision Letter 0]

13 Dec 2019

PONE-D-19-31205

Frailty predicts surgical complications after kidney transplantation. A propensity score matched study.

PLOS ONE

Dear Dr. Monteiro de Barros Almeida,

Thank you for submitting your manuscript to PLOS ONE. After careful consideration, we feel that it has merit but does not fully meet PLOS ONE’s publication criteria as it currently stands. Therefore, we invite you to submit a revised version of the manuscript that addresses the points raised during the review process.

Timely and important topic in the field of kidney transplantation. One statistical reviewer and two experts in the field have raised several issues on methodology and execution of the study, as well as interpretation of the results. These concerns will have to be adressed thoroughly. Also, quite a few clarifications are needed by the authors. Please make a real effort to revise the MS accordingly. This is no guarantee that the revised MS will be accepted, though.

We would appreciate receiving your revised manuscript by Jan 27 2020 11:59PM. To enhance the reproducibility of your results, we recommend that if applicable you deposit your laboratory protocols in protocols.io, where a protocol can be assigned its own identifier (DOI) such that it can be cited independently in the future. For instructions see: http://journals.plos.org/plosone/s/submission-guidelines#loc-laboratory-protocols

We look forward to receiving your revised manuscript.

Kind regards,

Frank JMF Dor, M.D., Ph.D., FEBS, FRCS

Academic Editor

PLOS ONE

Journal Requirements:

2. Please provide additional details regarding participant consent. In the ethics statement in the Methods and online submission information, please ensure that you have specified whether consent was suitably informed.

Reviewers' comments:

Reviewer's Responses to Questions

**Comments to the Author**

1. Is the manuscript technically sound, and do the data support the conclusions?

Reviewer #1: Partly

Reviewer #2: Partly

Reviewer #3: Partly

2. Has the statistical analysis been performed appropriately and rigorously? 

Reviewer #1: Yes

Reviewer #2: I Don't Know

Reviewer #3: Yes

3. Have the authors made all data underlying the findings in their manuscript fully available?

Reviewer #1: Yes

Reviewer #2: No

Reviewer #3: Yes

4. Is the manuscript presented in an intelligible fashion and written in standard English?

Reviewer #1: Yes

Reviewer #2: Yes

Reviewer #3: Yes

5. Review Comments to the Author

Reviewer #1: The manuscript entitled 'Frailty predicts surgical complications after kidney transplantation. A propensity score matched study. ' with the aim to determine of frailty are predictive of surgical complications following kidney transplant.

This is quite an interesting study. The manuscript can be further improved.

Comments

Methods

It would be good to have sample size calculation for the study.

Figure 1, more detail information to be provided to describe the study i.e. flow, grouping, M0, M1 etc

Statistical analysis

Statement on statistical analysis were performed on 'before matching' and 'after matching' to be added. 1 or 2 tailed for the statistical test(s) to be stated.

Results

Table 1, statistical test to be denoted in the table footnote.

Line 212- 220, the figure of RR, 95%CI to be placed in Table 3. 95% CI or 95% confidence interval (CI) to be standardized.

Table 3, individual n and total complications for both before and after matching to be added. Statistical test to be denoted in the table footnote.

Figure 2, 3, n (%) to be labelled in the graph for easy visualization.

Reviewer #2: Predicting complications and hence making the decision as to whether to proceed with transplantation is a complex process. There having been a growing number of publication looking at some for frailty index and this study adds to that body of work and hence is timely. The frailty methodology (Fried) used here is a well validated method of calculating frailty in renal patients.

There are some major questions that need clarification:

1. There were 131 transplants during this period but only 88 were included in the study. The area of concerns is that a total of 24 of the 44 exclusions were attributed to either researcher not informed or logistical difficulties. This needs further to be explained in more detail as there is a real risk of systematic bias.

2. It is stated that iatrogenic surgical complications were excluded from the analysis. It is not clear how it was determined if a complication was iatrogenic. Of note the commonest complication was urinary fistula. Under certain circumstances this could be considered an iatrogenic complication. Similarly it would be useful to know what complications were excluded from consideration as they were thought to be iatrogenic.

Reviewer #3: Thank you for the opportunity to review this paper which is of huge relevance and significance within the current transplant climate. I think it has been relatively well written and raises some interesting points.

My main issue with the study is that the measure used to assess frailty does not appear to have been validated for the renal failure population. The measure used (Fried et al) was written "To develop and operationalize a phenotype of frailty in older adults" and was validated based on a sample of patients who were all aged over 65. To use the same measure in the renal failure population awaiting transplantation is problematic for a number of reasons. Firstly, they are, on average, 20 years younger. Secondly, many of the sequela of ESRD are similar to that of frailty and therefore it is impossible to know whether the scores from the questionnaire are demonstrating true frailty or symptoms of ESRD. For example, weight loss is a difficult factor to consider within the dialysis population due to issues with fluid retention, whether patients are on a fluid restriction, whether they had been dialysed prior to admission and so on. It is therefore imperative that a measure of this kind is validated within the population in which you wish to use it.

With regard to the conclusions of the study, I agree that the finding of non-infective complications being at a higher rate in the more frail population is of interest, however I am unsure (given the reasons outlined above) how reliable this data is. Reason being, symptoms or signs that may be labelled as 'frailty' may actually be signs of something else related to the renal failure that would also have put the patient at higher risk of complications.

More minor points:

Abstract - I would expand the results and discussion sections. A single line for these parts of the abstract are not really adequate in my opinion

I would not class a lymphocele as a urological complication

I am unsure what a urinary fistula is - please explain what is meant by this

Exclusion criteria - you say that 'those whose physical or congnitive conditions prevented the calculation of the frailty score were excluded from the study' however you do not expand on what these are.

6. PLOS authors have the option to publish the peer review history of their article (what does this mean?). If published, this will include your full peer review and any attached files.

Reviewer #1: No

Reviewer #2: No

Reviewer #3: No

---

## [Author Response · Author response to Decision Letter 0]

5 Jan 2020

Dear Editor,

We really appreciate all the comments made about our manuscript. Below we answer the questions and suggestions made by the reviewers.

Dear reviewers,

Thank you very much for the comments. They were extremely helpful. We are available to answer any further questions or suggestions you might have.

Reviewer #1: The manuscript entitled 'Frailty predicts surgical complications after kidney transplantation. A propensity score matched study. ' with the aim to determine of frailty are predictive of surgical complications following kidney transplant.

This is quite an interesting study. The manuscript can be further improved.

Comments

Methods

It would be good to have sample size calculation for the study.

Answer: We provided the sample size calculation in ‘Statistical analysis’ section.

Figure 1, more detail information to be provided to describe the study i.e. flow, grouping, M0, M1 etc 

Answer: Done as suggested. Please see the new figure 1 attached.

Statistical analysis

Statement on statistical analysis were performed on 'before matching' and 'after matching' to be added. 1 or 2 tailed for the statistical test(s) to be stated.

Answer: Done as suggested.

Results

Table 1, statistical test to be denoted in the table footnote.

Answer: Statistical tests were denoted in the Table 1 footnote.

Line 212- 220, the figure of RR, 95%CI to be placed in Table 3. 95% CI or 95% confidence interval (CI) to be standardized.

Answer: Confidence interval was standardized in the text. The 95% CI was provided in Table 3.

Table 3, individual n and total complications for both before and after matching to be added. Statistical test to be denoted in the table footnote.

Answer: Done as suggested.

Figure 2, 3, n (%) to be labelled in the graph for easy visualization. 

Answer: We modified the figures 2 and 3 for easy visualization.

Reviewer #2: Predicting complications and hence making the decision as to whether to proceed with transplantation is a complex process. There having been a growing number of publication looking at some for frailty index and this study adds to that body of work and hence is timely. The frailty methodology (Fried) used here is a well validated method of calculating frailty in renal patients.

There are some major questions that need clarification:

1. There were 131 transplants during this period but only 88 were included in the study. The area of concerns is that a total of 24 of the 44 exclusions were attributed to either researcher not informed or logistical difficulties. This needs further to be explained in more detail as there is a real risk of systematic bias.

Answer: We combined the terms ‘researcher not informed’, ‘logistical difficulties’, and ‘no available time for collection’ into a single term named ‘logistical difficulties’ because they represent similar reasons for non-inclusion in the study. Logistical difficulties occurred mainly due to communication difficulties between the transplantation team and the research team at patient admission. However, we believe that this potential bias does not represent a relevant risk, as it is not possible to identify patients as frail or non-frail without applying the methodology of Fried et al., nor can we anticipate kidney transplant outcomes at the time patients are hospitalized. We added a statement in the ‘Results’ section that explains in more detail the logistical difficulties.

2. It is stated that iatrogenic surgical complications were excluded from the analysis. It is not clear how it was determined if a complication was iatrogenic. Of note the commonest complication was urinary fistula. Under certain circumstances this could be considered an iatrogenic complication. Similarly it would be useful to know what complications were excluded from consideration as they were thought to be iatrogenic.

Answer: We have initially proposed to define ‘iatrogenic surgical complications’ as organ and tissue injuries that accidentally occur at the surgical site during the procedure. However, we agree with the reviewer that the definition of ‘iatrogenic surgical complications’ is rather inaccurate. That’s why we removed this statement from the manuscript. Additionally, no patients were excluded due to this condition. 

Reviewer #3: Thank you for the opportunity to review this paper which is of huge relevance and significance within the current transplant climate. I think it has been relatively well written and raises some interesting points.

My main issue with the study is that the measure used to assess frailty does not appear to have been validated for the renal failure population. The measure used (Fried et al) was written "To develop and operationalize a phenotype of frailty in older adults" and was validated based on a sample of patients who were all aged over 65. To use the same measure in the renal failure population awaiting transplantation is problematic for a number of reasons. Firstly, they are, on average, 20 years younger. Secondly, many of the sequela of ESRD are similar to that of frailty and therefore it is impossible to know whether the scores from the questionnaire are demonstrating true frailty or symptoms of ESRD. For example, weight loss is a difficult factor to consider within the dialysis population due to issues with fluid retention, whether patients are on a fluid restriction, whether they had been dialysed prior to admission and so on. It is therefore imperative that a measure of this kind is validated within the population in which you wish to use it.

Answer: Frailty has been shown to be applicable in end-stage renal disease patients (ESRD) and in kidney-transplanted patients. These populations appear to share many pathogenic mechanisms of frailty, such as a pro-inflammatory state, with an exacerbated production of inflammatory cytokines, and dysregulation of the immune, neuroendocrine, and neuromuscular systems, resulting in accelerated ageing [1-4]. Frailty is considered highly prevalent in patients at any stage of chronic kidney disease (CKD) and may reach up to two-thirds in ESRD cases [5]. The use of the frailty score has progressively increased in the ESRD and transplant population. A recent systematic review [6] evaluating the use of the frailty score in the population of patients with CKD found that 72% of the studies used the Fried et al. score.

In kidney transplantation, frailty has been shown to be an independent risk factor for delayed graft function [1], post-transplant delirium [7], longer initial length of hospital stay [8], early hospital readmission [9], adverse effects from immunosuppressive drugs [10], and death [11]. In the ‘Report from the American Society of Transplantation on frailty in solid organ transplantation’, 98.9% of transplant surgeons responded that frailty in transplant candidates is a risk factor for adverse outcomes after transplantation [12].

We think that the measure of weight loss when calculating the Fried score in dialysis population was not a significant limitation because we only considered self-report of unintentional dry weight loss, reducing this limitation. Beside that, the Fried score has been used as a robust marker of outcomes in ESRD patients.

We added the two following statements in the ‘Introduction’ and ‘Material and methods’ sections, respectively, to emphasize the topics discussed above:

- ‘Frailty is considered highly prevalent in patients at any stage of chronic kidney disease (CKD), and may reach up to two-thirds in ESRD cases [Kojima G. Prevalence of frailty in end-stage renal disease: a systematic review and meta-analysis. Int Urol Nephrol. 2017;49 (11):1989–1997]. The use of the frailty score has progressively increased in the ESRD and transplant population. A recent systematic review [Chowdhury R, Peel NM, Krosch M, Hubbard RE. Frailty and chronic kidney disease: A systematic review. Arch Gerontol Geriatr. 2017;68:135–142] evaluating the use of the frailty score in the population of patients with CKD found that 72% of the studies used the Fried et al. score.’

- ‘Fried et al. score was calculated considering only the dry weight loss.’

References 

1. Garonzik-Wang JM, Govindan P, Grinnan JW, Liu M, Ali HM, Chakraborty A, et al. Frailty and delayed graft function in kidney transplant recipients. Arch Surg. 2012;147(2):190-193.

2. Exterkate L, Slegtenhorst BR, Kelm M, Seyda M, Schuitenmaker JM, Quante M, et al. Frailty and transplantation. Transplantation. 2016;100(4):727-733.

3. McAdams-DeMarco MA, Suresh S, Law A, Salter ML, Gimenez LF, Jaar BG, et al. Frailty and falls among adult patients undergoing chronic hemodialysis: A prospective cohort study. BMC Nephrol. 2013;214:224. 

4. McAdams‐DeMarco MA, Law A, Salter ML, Boyarsky B, Gimenez L, Jaar BG, et al. Frailty as a novel predictor of mortality and hospitalization in individuals of all ages undergoing hemodialysis. J Am Geriatr Soc. 2013;61(6):896-901.

5. Kojima G. Prevalence of frailty in end-stage renal disease: a systematic review and meta-analysis. Int Urol Nephrol. 2017;49(11):1989–1997.

6. Chowdhury R, Peel NM, Krosch M, Hubbard RE. Frailty and chronic kidney disease: A systematic review. Arch Gerontol Geriatr. 2017;68:135–142.

7. Haugen CE, Mountford A, Warsame F, Berkowitz R, Bae S, Thomas AG, et al. Incidence, risk factors, and sequelae of post‐kidney transplant delirium. J Am Soc Nephrol. 2018;29(6):1752-1759.

8. McAdams‐DeMarco MA, King EA, Luo X, Haugen C, DiBrito S, Shaffer A, et al. Frailty, length of stay, and mortality in kidney transplant recipients: a national registry and prospective cohort study. Ann Surg. 2016;266(6):1084‐1090.

9. McAdams-DeMarco MA, Law A, Salter ML, Chow E, Grams M, Walston J, et al. Frailty and early hospital readmission after kidney transplantation. Am J Transplant. 2013;13(8):2091-2095.

10. McAdams-De Marco MA, Law A, Tan J, Delp C, King EA, Orandi, et al. Frailty, mycophenolate reduction, and graft loss in kidney transplant recipients. Transplantation. 2015;15(1):149-154.

11. Mc Adams-De Marco MA, Law A, King E, Orandi B, Salter M, Gupta N, et al. Frailty and mortality in Kidney transplant recipients. Am J Transplant. 2015;15(1):149-154.

12. Kobashigawa J, Dadhania D, Bhorade S, Adey D, Berger J, Bhat G, et al. Report from the American Society of Transplantation on frailty in solid organ transplantation. Am J Transplant. 2019;19(4): 984-994.

With regard to the conclusions of the study, I agree that the finding of non-infective complications being at a higher rate in the more frail population is of interest, however I am unsure (given the reasons outlined above) how reliable this data is. Reason being, symptoms or signs that may be labelled as 'frailty' may actually be signs of something else related to the renal failure that would also have put the patient at higher risk of complications.

Answer: According to the discussion and the literature above, we believe that Fried et al. frailty score is indeed applicable to ESRD and transplant populations. Frailty score can be a robust marker that predicts important outcomes after transplantation. The most important is that the Fried et al. score can capture higher risk patients in kidney transplantation.

More minor points:

Abstract - I would expand the results and discussion sections. A single line for these parts of the abstract are not really adequate in my opinion

Answer: We agree with the reviewer. The results and discussion sections in the abstract were expanded.

I would not class a lymphocele as a urological complication

Answer: Some references classified the lymphocele as vascular and others as urological complication. We agree to reclassify lymphocele as a vascular complication.

I am unsure what a urinary fistula is - please explain what is meant by this

Answer: We refered urinary leak as urinary fistula. We changed the terms in the text.

Exclusion criteria - you say that 'those whose physical or congnitive conditions prevented the calculation of the frailty score were excluded from the study' however you do not expand on what these are.

Answer: We changed the phrase to ‘Individuals with amputations or other physical conditions that prevented the walk test or the handgrip strength test and the patients with significant cognitive impairment who were unable to understand and respond to the frailty score questionnaires were also excluded from the study’.

---

## [Decision Letter · Decision Letter 1]

30 Jan 2020

PONE-D-19-31205R1

Frailty predicts surgical complications after kidney transplantation. A propensity score matched study.

PLOS ONE

Dear Dr. Monteiro de Barros Almeida,

Thank you for submitting your manuscript to PLOS ONE. After careful consideration, we feel that it has merit but does not fully meet PLOS ONE’s publication criteria as it currently stands. Therefore, we invite you to submit a revised version of the manuscript that addresses the points raised during the review process.

ACADEMIC EDITOR: 

The authors have addressed most of the point raised by the reviewers, but there are still a few outstanding points that would warrant careful revision. Please see detailed responses by the three reviewers.

We would appreciate receiving your revised manuscript by Mar 15 2020 11:59PM. To enhance the reproducibility of your results, we recommend that if applicable you deposit your laboratory protocols in protocols.io, where a protocol can be assigned its own identifier (DOI) such that it can be cited independently in the future. For instructions see: http://journals.plos.org/plosone/s/submission-guidelines#loc-laboratory-protocols

We look forward to receiving your revised manuscript.

Kind regards,

Frank JMF Dor, M.D., Ph.D., FEBS, FRCS

Academic Editor

PLOS ONE

Reviewers' comments:

Reviewer's Responses to Questions

**Comments to the Author**

1. If the authors have adequately addressed your comments raised in a previous round of review and you feel that this manuscript is now acceptable for publication, you may indicate that here to bypass the “Comments to the Author” section, enter your conflict of interest statement in the “Confidential to Editor” section, and submit your "Accept" recommendation.

Reviewer #1: All comments have been addressed

Reviewer #2: (No Response)

Reviewer #3: All comments have been addressed

2. Is the manuscript technically sound, and do the data support the conclusions?

Reviewer #1: (No Response)

Reviewer #2: Partly

Reviewer #3: Yes

3. Has the statistical analysis been performed appropriately and rigorously? 

Reviewer #1: (No Response)

Reviewer #2: I Don't Know

Reviewer #3: Yes

4. Have the authors made all data underlying the findings in their manuscript fully available?

Reviewer #1: (No Response)

Reviewer #2: Yes

Reviewer #3: Yes

5. Is the manuscript presented in an intelligible fashion and written in standard English?

Reviewer #1: (No Response)

Reviewer #2: Yes

Reviewer #3: Yes

6. Review Comments to the Author

Reviewer #1: Minor comments

Page 7 Line 161, alfa to be written as alpha.

Page 7 Line 162, to state clearly sample size 56 for whole group or each group. Also to add discussion on power of study.

Page 12 Table 3, CI 95% to be written as 95%CI.

Reviewer #2: I raised 2 questions when originally reviewing this paper. The authors have answered question 2 satisfactorily in my opinion. As for question 1, they have partly addressed my concerns but not fully. I think it is worth their including a statement raising the possibility that the exclusion of patients due to logicistical difficulties may have resulted in a systematic bias. Perhaps the clinical team did not try as hard to contact the Research team for a certain "type" of patient. I think a statement addressing this and a comment of why they do not think it relevant would satisfy me. If the editors are happy with the changes made I do not need to see the manuscript again

Reviewer #3: Thank you for providing a detailed response to my comments and for making appropriate adjustments to the manuscipt.

My main comment is with regard to the statement about questionnaire validation. I think it is entirely reasonable to make a statement saying that frailty is being increasingly assessed within this population and that the Fried score appears to be the most popular rating scale being used. However, I still think there also needs to be a statement that says something akin to "The Fried frailty index has not been formally validated within the end-stage renal failure population, however it is currently the most popular / frequently used questionnaire".

This is important because the reader needs to be aware that whilst this is the most popular scale, it has not undergone the formal validation process that a lot of questionnaires should have done when being used in certain populations. The crossover of symptoms between those with frailty due to renal disease and general frailty is significant and the fact that this has not specifically been considered as part of validity testing is hugely relevant within this context.

7. PLOS authors have the option to publish the peer review history of their article (what does this mean?). If published, this will include your full peer review and any attached files.

Reviewer #1: No

Reviewer #2: No

Reviewer #3: No

---

## [Author Response · Author response to Decision Letter 1]

3 Feb 2020

Dear Editor,

We sincerely appreciate all valuable comments and suggestions. We have made a great effort to take into account all the suggestions proposed by the reviewers. We are available to answer any further questions or suggestions you might have.

Reviewer #1: Minor comments.

Page 7 Line 161, alfa to be written as alpha.

Answer: Thank you very much. We changed to “alpha”.

Page 7 Line 162, to state clearly sample size 56 for whole group or each group. Also to add discussion on power of study. 

Answer: The reviewer is correct. We apologize about the mistake. In fact, we assumed a frequency of surgical complications of 45% in the frail group and 15% in the non-frail group. Considering a power of 0.80, and alpha of 0.05, we calculated a minimum sample size of 68 patients. This information was corrected in the ‘Statistical analysis’ section. We also calculated the post hoc power of the study considering the incidence of surgical complications after matching and it reached 0.71. However, it is important to notice that there was a significant difference between the groups, thus type I error and not type II is the most important in our study.

Page 12 Table 3, CI 95% to be written as 95%CI.

Answer: Done as suggested.

Reviewer #2: I raised 2 questions when originally reviewing this paper. The authors have answered question 2 satisfactorily in my opinion. As for question 1, they have partly addressed my concerns but not fully. I think it is worth their including a statement raising the possibility that the exclusion of patients due to logicistical difficulties may have resulted in a systematic bias. Perhaps the clinical team did not try as hard to contact the Research team for a certain "type" of patient. I think a statement addressing this and a comment of why they do not think it relevant would satisfy me. If the editors are happy with the changes made I do not need to see the manuscript again

Answer: We added the following statement in the ‘Discussion’ section: ‘The exclusion of patients due to logistical difficulties may have resulted in a systematic bias. However, we believe that this potential bias does not represent a relevant risk, as we cannot anticipate the frail condition without applying the Fried score [13]’.

Reviewer #3: Thank you for providing a detailed response to my comments and for making appropriate adjustments to the manuscipt.

My main comment is with regard to the statement about questionnaire validation. I think it is entirely reasonable to make a statement saying that frailty is being increasingly assessed within this population and that the Fried score appears to be the most popular rating scale being used. However, I still think there also needs to be a statement that says something akin to "The Fried frailty index has not been formally validated within the end-stage renal failure population, however it is currently the most popular / frequently used questionnaire".

This is important because the reader needs to be aware that whilst this is the most popular scale, it has not undergone the formal validation process that a lot of questionnaires should have done when being used in certain populations. The crossover of symptoms between those with frailty due to renal disease and general frailty is significant and the fact that this has not specifically been considered as part of validity testing is hugely relevant within this context.

Answer: Thank you very much for the comments. In the ‘Introduction’ section we changed the phrase ‘Frailty is a measure of physiological reserve, initially validated for the geriatric population [13]. However, it has been shown to be applicable in end-stage renal disease (ESRD) patients and in kidney-transplanted patients. These populations appear to share many pathogenic mechanisms of frailty, such as a pro-inflammatory state, with an exacerbated production of inflammatory cytokines, and dysregulation of the immune, neuroendocrine, and neuromuscular systems, resulting in accelerated ageing [12,14-16]’ to ‘Frailty is a measure of physiological reserve, initially validated for the geriatric population [13]. Although the frailty score has not been formally validated for patients with end-stage renal disease (ESRD) and for kidney-transplanted patients, it has been shown to be applicable to these populations. These patients appear to share many pathogenic mechanisms of frailty, such as a pro-inflammatory state, with an exacerbated production of inflammatory cytokines, and dysregulation of the immune, neuroendocrine, and neuromuscular systems, resulting in accelerated ageing [12,14-16]’.

We think this statement has clarified the fact that frailty score has not yet undergone the formal validation process within ESRD and kidney-transplanted populations.

---

## [Decision Letter · Decision Letter 2]

10 Feb 2020

Frailty predicts surgical complications after kidney transplantation. A propensity score matched study.

PONE-D-19-31205R2

Dear Dr. Monteiro de Barros Almeida,

We are pleased to inform you that your manuscript has been judged scientifically suitable for publication and will be formally accepted for publication once it complies with all outstanding technical requirements.

With kind regards,

Frank JMF Dor, M.D., Ph.D., FEBS, FRCS

Academic Editor

PLOS ONE

Additional Editor Comments (optional):

Reviewers' comments:

Reviewer's Responses to Questions

**Comments to the Author**

1. If the authors have adequately addressed your comments raised in a previous round of review and you feel that this manuscript is now acceptable for publication, you may indicate that here to bypass the “Comments to the Author” section, enter your conflict of interest statement in the “Confidential to Editor” section, and submit your "Accept" recommendation.

Reviewer #1: All comments have been addressed

Reviewer #2: All comments have been addressed

Reviewer #3: All comments have been addressed

2. Is the manuscript technically sound, and do the data support the conclusions?

Reviewer #1: (No Response)

Reviewer #2: Yes

Reviewer #3: Yes

3. Has the statistical analysis been performed appropriately and rigorously? 

Reviewer #1: (No Response)

Reviewer #2: I Don't Know

Reviewer #3: Yes

4. Have the authors made all data underlying the findings in their manuscript fully available?

Reviewer #1: (No Response)

Reviewer #2: Yes

Reviewer #3: Yes

5. Is the manuscript presented in an intelligible fashion and written in standard English?

Reviewer #1: Yes

Reviewer #2: Yes

Reviewer #3: Yes

6. Review Comments to the Author

Reviewer #1: (No Response)

Reviewer #2: (No Response)

Reviewer #3: All my comments have now been addressed, thank you.

7. PLOS authors have the option to publish the peer review history of their article (what does this mean?). If published, this will include your full peer review and any attached files.

Reviewer #1: No

Reviewer #2: No

Reviewer #3: No

---

## [Editor Report · Acceptance letter]

13 Feb 2020

PONE-D-19-31205R2 

Frailty predicts surgical complications after kidney transplantation. A propensity score matched study. 

Dear Dr. Monteiro de Barros Almeida:

I am pleased to inform you that your manuscript has been deemed suitable for publication in PLOS ONE. Congratulations! Your manuscript is now with our production department. 

With kind regards,

on behalf of

Dr. Frank JMF Dor 

Academic Editor

PLOS ONE